# Procalcitonin for antimicrobial stewardship among cancer patients admitted with COVID-19

Hiba Dagher[1], Anne-Marie Chaftari[1]*, Patricia Mulanovich[1], Ying Jiang[1], Ray Hachem[1], Alexandre E Malek[1], Jovan Borjan[2], George M Viola[1], Issam Raad[1]

[1]Department of Infectious Diseases, Infection Control and Employee Health, The University of Texas MD Anderson Cancer Center, Houston, United States; [2]Pharmacy Clinical Programs, The University of Texas MD Anderson Cancer Center, Houston, United States

*For correspondence:
achaftari@mdanderson.org

Competing interest: The authors declare that no competing interests exist.

## Abstract

**Background:** Procalcitonin (PCT) has been used to guide antibiotic therapy in bacterial infections. We aimed to determine the role of PCT in decreasing the duration of empiric antibiotic therapy among cancer patients admitted with COVID-19.

**Methods:** This retrospective study included cancer patients admitted to our institution for COVID-19 between March 1, 2020, and June 28, 2021, with a PCT test done within 72 hr after admission. Patients were divided into two groups: PCT <0.25 ng/ml and PCT ≥0.25 ng/ml. We assessed pertinent cultures, antibacterial use, and duration of empiric antibacterial therapy.

**Results:** The study included 530 patients (median age, 62 years [range, 13–91]). All the patients had ≥1 culture test within 7 days following admission. Patients with PCT <0.25 ng/ml were less likely to have a positive culture than were those with PCT ≥0.25 ng/ml (6% [20/358] vs. 17% [30/172]; p<0.0001). PCT <0.25 ng/ml had a high negative predictive value for bacteremia and 30 day mortality. Patients with PCT <0.25 ng/ml were less likely to receive intravenous (IV) antibiotics for >72 hr than were patients with PCT ≥0.25 ng/ml (45% [162/358] vs. 69% [119/172]; p<0.0001). Among patients with PCT <0.25 ng/ml and negative cultures, 30 day mortality was similar between those who received IV antibiotics for ≥72 hr and those who received IV antibiotics for shorter durations (2% [2/111] vs. 3% [5/176], p=0.71).

**Conclusions:** Among cancer patients with COVID-19, PCT level <0.25 ng/ml is associated with lower likelihood of bacterial co-infection and greater likelihood of a shorter antibiotic course. In patients with PCT level <0.25 ng/ml and negative cultures, an antibiotic course of >72 hr may not be necessary. PCT could be useful in enhancing antimicrobial stewardship in cancer patients with COVID-19.

**Funding:** This research was supported by the National Institutes of Health/National Cancer Institute under award number P30CA016672, which supports MD Anderson Cancer Center's Clinical Trials Office.

## Editor's evaluation

One must appreciate the challenges of antimicrobial stewardship in an immunocompromised population. This retrospective single-institution study provides valuable support for the working hypothesis that initial procalcitonin levels might be used in cancer patients admitted with COVID-19 infection to omit, reduce, or de-escalate the need for empiric antimicrobial therapy. In the setting of a global pandemic, this is a common issue with COVID-19 patients in general, but far more difficult in a cancer patient population. The results presented are solid, however, future subgroup analysis

of more specific scenarios among cancer patients with COVID-19 (e.g., neutropenia, active chemo-therapy, and need for intensive care) are warranted.

## Introduction

Many factors predicting the outcome and prognosis of coronavirus disease 2019 (COVID-19) have been identified. These factors have proved valuable for determining prognosis and have guided the treatment of patients at risk for severe COVID-19. Procalcitonin (PCT) is a biomarker that has served as an indicator for bloodstream infections and has been used as a guide to antimicrobial management in sepsis and bacterial infections in the general population (*ElGohary et al., 2022*; *Azzini et al., 2020*; *Kalil et al., 2016*; *Schuetz et al., 2013*) and in cancer patients with and without neutropenia (*Haddad et al., 2018*; *El Haddad et al., 2018*; *Chaftari et al., 2021b*; *Chaftari et al., 2021a*). Randomized trials have shown PCT to be useful in guiding decisions regarding antimicrobial therapy for lower respiratory tract infections (*Schuetz et al., 2017*; *Schuetz et al., 2009*; *Christ-Crain et al., 2006*). Several PCT cut-off values have been evaluated and used in different treatment algorithms. PCT cut-off values of 0.25 and 0.5 ng/ml have been adopted for critically ill patients in the intensive care unit (ICU) (*Bouadma et al., 2010*), neutropenic patients (*Azzini et al., 2020*; *Kalil et al., 2016*; *Chaftari et al., 2021a*), and patients with lower respiratory tract infections (*Christ-Crain et al., 2004*).

In patients with coronavirus disease 2019 (COVID-19), elevated PCT levels and elevated levels of other inflammatory markers have been associated with more severe COVID-19 both in the general population (*Haddad et al., 2018*; *Frater et al., 2020*; *Pink et al., 2021*; *Ponti et al., 2020*; *Lippi and Plebani, 2020*) and in cancer patients (*ElGohary et al., 2022*; *Cai et al., 2021*).

Bacterial co-infections may not be prevalent in patients with COVID-19 (*Rawson et al., 2020*). However, because of the similarity in signs and symptoms between bacterial co-infections and COVID-19 and the difficulty of ruling out a bacterial infection in patients presenting with COVID-19 pneumonia, empirical treatment with antibiotics is often initiated in patients with COVID-19 without a confirmed bacterial co-infection (*Rawson et al., 2020*). This practice may lead to an emergence of antibiotic resistance, undesirable adverse events, and increase costs (*Azzini et al., 2020*; *Kalil et al., 2016*). One study showed that the use of antibiotics in patients with COVID-19 with a PCT level >0.25 ng/ml and with a low suspicion of bacterial infection did not improve clinical outcome (*So et al., 2022*). Little to no data have been published regarding PCT for antimicrobial stewardship among cancer patients with COVID-19.

Given the widespread use of empiric antibiotics in cancer patients admitted for COVID-19, we evaluated the role of PCT in decreasing the duration of empiric antibiotic therapy in this patient population.

## Methods

We conducted a retrospective study of cancer patients who were admitted to The University of Texas MD Anderson Cancer Center between March 1, 2020, and June 28, 2021, for COVID-19 and the highest serum PCT level measured within 72 hr after admission was collected for the study. Patients were divided into two groups: PCT level <0.25 ng/ml and PCT level ≥0.25 ng/ml. This cut-off is conventionally suggested and has been used in different algorithms (*Christ-Crain et al., 2006*; *Bouadma et al., 2010*; *Christ-Crain et al., 2004*).

We reviewed the patients' electronic medical records and collected data pertinent to demographics (age, sex, and race and ethnicity), type of cancer (hematological malignancy vs. solid tumor), cancer status (active vs. in remission), active cancer therapy, co-morbidities, tobacco use, and presence of pneumonia. We assessed laboratory test results, including absolute neutrophil count, PCT level, documented bacterial infections, and sources of cultures. We also extracted data on oxygen saturation, requirement for oxygen supplementation, need for and duration of intravenous (IV) antibiotic therapy, ICU admission, and 30 day mortality after COVID-19 diagnosis. Pneumonia was defined as an abnormal chest imaging (chest radiograph or computed tomography scan) in patients who present with respiratory symptoms.

Our study was approved by the Institutional Review Board of MD Anderson Cancer Center, and a waiver of informed consent was obtained.

**Table 1.** Baseline patient characteristics of hospitalized cancer patients with coronavirus disease 2019 (COVID-19) with different procalcitonin (PCT) levels[*].

| Characteristic | PCT <0.25 ng/ml | PCT ≥0.25 ng/ml | p-Value |
|---|---|---|---|
| | (n=358) | (n=172) | |
| Age, median (range), years | 61 (13–91) | 64 (14–86) | 0.11 |
| Sex, male | 178 (50) | 95 (55) | 0.23 |
| Type of cancer | | | 0.79 |
| Hematological malignancy only | 142 (40) | 63 (37) | |
| Solid tumor only | 195 (54) | 99 (58) | |
| Both of above | 21 (6) | 10 (6) | |
| Status of cancer | | | 0.90 |
| Active | 315 (88) | 152 (88) | |
| No evidence of disease | 43 (12) | 20 (12) | |
| Active cancer therapy within 30 days | 118 (33) | 58 (34) | 0.86 |
| Chemotherapy received | 272 (76) | 121 (70) | 0.16 |
| Chronic kidney disease | 114/327 (35) | 89/168 (53) | <0.001 |
| Asthma | 44/327 (13) | 23/168 (14) | 0.94 |
| Chronic obstructive pulmonary disease | 59/327 (18) | 34/168 (20) | 0.55 |
| Congestive heart failure | 46/327 (14) | 33/168 (20) | 0.11 |
| Diabetes mellitus | 164/327 (50) | 83/168 (49) | 0.87 |
| Coronary artery disease | 12/327 (4) | 3/168 (2) | 0.25 |
| Hypertension | 251/327 (77) | 137/168 (82) | 0.22 |
| Venous thromboembolic event | 42/327 (13) | 19/168 (11) | 0.62 |
| Obesity | 37/327 (11) | 20/168 (12) | 0.85 |
| Obstructive sleep apnea | 55/327 (17) | 18/168 (11) | 0.07 |

[*]Values in table are number of patients (percentage) unless otherwise indicated.

We compared the clinical characteristics and outcomes of the patients in the PCT <0.25 ng/ml and PCT ≥0.25 ng/ml groups. We used the $\chi^2$ or Fisher's exact test, as appropriate, to compare categorical variables. We used Wilcoxon rank-sum tests to compare continuous variables because of the deviation of the data from the normal distribution. In addition, we used multivariable logistic regression model to evaluate the independent impact of PCT >0.25 ng/ml on each outcome we evaluated. We assessed negative predictive values of PCT levels for the prediction of the various outcomes. We also estimated the relative risks (RR) of various outcomes for a patient with PCT ≥0.25 ng/ml. All tests were two-sided at a significance level of 0.05. The statistical analyses were performed using SAS version 9.3 (SAS Institute Inc, Cary, NC).

## Results

We identified 530 patients, of whom 358 (68%) had a PCT level <0.25 ng/ml and 172 (32%) had a PCT level ≥0.25 ng/ml. Patients in the two PCT groups were similar in terms of age, sex, race and ethnicity, type and status of cancer, and active cancer therapy (*Table 1*). The proportion of patients with an absolute neutrophil count <1000/µl was 9% in both groups; however, the proportion of patients with an absolute lymphocyte count <1000/µl was lower in patients with PCT <0.25 ng/ml (63% vs. 75%; p=0.009) (*Table 2*).

**Table 2.** Characteristics of COVID hospital admission in cancer patients with coronavirus disease 2019 (COVID-19) with different procalcitonin (PCT) levels[*].

| | PCT <0.25 ng/ml | PCT ≥0.25 ng/ml | p-Value |
|---|---|---|---|
| | (n=358) | (n=172) | |
| ANC <1000/μl at admission | 33/352 (9) | 16 (9) | 0.98 |
| ALC <1000/μl at admission | 219/348 (63) | 126/169 (75) | 0.009 |
| Pneumonia | 270/357 (76) | 141 (82) | 0.10 |
| Oxygen supplementation within 72 hr | 191/356 (54) | 118 (69) | 0.001 |
| Positive bacterial culture | 20 (6) | 30 (17) | <0.0001 |
| Site of positive culture | | | |
| Blood | 2/20 (10) | 10/30 (33) | 0.09 |
| Lower respiratory tract | 5/20 (25) | 10/30 (33) | 0.53 |
| Wound | 6/20 (30) | 5/30 (17) | 0.31 |
| Urine | 10/20 (50) | 8/30 (27) | 0.09 |
| Transfusion reaction culture | 0/20 (0) | 1/30 (3) | >0.99 |
| Cerebrospinal fluid | 0/20 (0) | 1/30 (3) | >0.99 |
| Positive fungal culture | 3 (1) | 8 (5) | 0.007 |
| Viral co-infection | 2 (1) | 0 (0) | >0.99 |
| Duration of hospital stay, median (IQR), days | 6 (4–10) | 10 (6–18) | <0.0001 |
| IV antibiotic treatment | 271 (76) | 154 (90) | <0.001 |
| Duration of IV antibiotic treatment, median (IQR), days | 4 (2–6) | 6 (3–7) | <0.0001 |
| Duration of IV antibiotic therapy ≥72 hr | 162 (45) | 119 (69) | <0.0001 |
| Duration of IV antibiotic therapy ≥7 days | 54 (15) | 54 (31) | <0.0001 |
| ICU admission | 51 (14) | 50 (29) | <0.0001 |
| Duration of ICU stay, median (IQR), days | 1 (1–4) | 3 (1–3) | 0.13 |
| Mortality within 30 days of COVID-19 diagnosis | 20 (6) | 33 (19) | <0.0001 |

[*]Values in table are number of patients (percentage) unless otherwise indicated.
ALC = absolute lymphocyte count. ANC = absolute neutrophil count. IQR = interquartile range.

Patients with PCT <0.25 ng/ml were less likely to require oxygen supplementation within 72 hr of admission (54% vs. 69%, p=0.001); were less likely to have a positive bacterial culture (6% vs. 17%; p<0.0001) from any source, including blood, lower respiratory tract, and urine; and had a lower proportion of patients with pneumonia, although the difference was not significant (76% vs. 82%; p=0.10). Patients with PCT <0.25 ng/ml were less likely to receive IV antibiotic therapy than were patients with PCT ≥0.25 ng/ml (76% vs. 90%; p<0.001). Furthermore, patients with PCT <0.25 ng/ml had a shorter median duration of IV antibiotic therapy (4 days vs. 6 days; p<0.0001) and were less likely to receive antibiotics for ≥72 hr compared to patients with PCT ≥0.25 (45% vs. 69%; p<0.0001) (*Table 2*). Similar results were found among patients with negative culture results: those with PCT <0.25 ng/ml were less likely to receive IV antibiotics for ≥72 hr than those with PCT ≥0.25 ng/ml (44% vs. 67%; p<0.0001). In addition, among patients with PCT <0.25 ng/ml and negative culture results, those who received a long course of IV antibiotics (≥72 hr) and those who received a shorter course had similar 30 day mortality rates (2% vs. 3%, p=0.71) (*Table 3*). Compared to patients with PCT ≥0.25 ng/ml, patients with PCT <0.25 ng/ml had shorter median duration of hospital stay (6 days vs. 10 days; p<0.0001), lower rate of ICU admission (14% vs. 29%; p<0.0001), and lower rate of mortality within 30 days of COVID diagnosis (6% vs. 19%; p<0.0001). By subset data analyses, we also found

**Table 3.** Treatment and outcomes of hospitalized cancer patients with coronavirus disease 2019 (COVID-19) with procalcitonin (PCT) <0.25 ng/ml and negative bacterial cultures.

| Outcomes | Duration of antibiotic treatment | | p-Value |
|---|---|---|---|
| | <72 hr | ≥72 hr | |
| | (n=176) | (n=111) | |
| | N (%) | N (%) | |
| Duration of hospital stay (days), median (IQR) | 5 (3–7) | 7 (5–11) | <0.0001 |
| Mortality within 30 days of COVID-19 diagnosis | 5 (3) | 2 (2) | 0.71 |

Note: Patients with intensive care unit (ICU) admission during hospitalization and patients who died within 3 days after hospital admission were excluded from analysis.

similar significant associations between PCT level and outcomes among patients under active cancer treatment (*Supplementary file 2*), but not among patients with ANC <1000/μl at admission. However, we need to point it out that the latter subset analyses were limited by a low statistical power due to the small sample size. Furthermore, multivariable logistic regression analyses showed that PCT >0.25 ng/ml was an independent predictor of every outcome we evaluated after adjusting for the possible confounders (*Supplementary file 1*).

We also evaluated the negative predictive values of PCT <0.25 ng/ml for different outcomes. Analyses showed that PCT <0.25 ng/ml had a high negative predictive value for bacteremia (NPV = 0.94, 95% CI = 0.92–0.97) and 30 day mortality (NPV = 0.94, 95% CI = 0.92–0.97), followed by ICU admission (NPV = 0.86, 95% CI = 0.82–0.89) and IV antibiotic use ≥7 days (NPV = 0.85, 95% CI = 0.81–0.88) (*Table 4*). Correspondingly, PCT level ≥0.25 ng/ml was associated with elevated RR for 30 day mortality (RR = 3.43, 95% CI = 2.03–5.80) followed by positive bacterial culture (RR = 3.12, 95% CI = 1.83–5.34), IV antibiotic use ≥7 days (RR = 2.08, 95% CI = 1.50–2.90), and ICU admission (RR = 2.04, 95% CI = 1.45–2.88) (*Table 4*).

## Discussion

In this study of cancer patients admitted for COVID-19, we found that PCT level <0.25 ng/ml was associated with a lower rate of bacterial co-infection, shorter hospital stay, shorter duration of IV antibiotics, and lower 30 day mortality. We also found that among the patients with PCT <0.25 ng/

**Table 4.** NPV of PCT <0.25 ng/ml and relative risk (RR) associated with PCT ≥0.25 ng/ml for selected outcomes in hospitalized cancer patients with coronavirus disease 2019 (COVID-19).

| Outcome | NPV of PCT <0.25 ng/ml | 95% CI | RR of PCT ≥0.25 ng/ml | 95% CI |
|---|---|---|---|---|
| Positive bacterial culture | 0.94 | 0.92–0.97 | 3.12 | 1.83–5.34 |
| Use of IV antibiotics | 0.24 | 0.20–0.29 | 1.18 | 1.09–1.28 |
| Use of IV antibiotics ≥72 hr | 0.55 | 0.49–0.60 | 1.53 | 1.31–1.78 |
| Use of IV antibiotics ≥7 days | 0.85 | 0.81–0.88 | 2.08 | 1.50–2.90 |
| ICU admission | 0.86 | 0.82–0.89 | 2.04 | 1.45–2.88 |
| Death within 30 days after COVID-19 diagnosis | 0.94 | 0.92–0.97 | 3.43 | 2.03–5.80 |

NPV = negative predictive value. RR = relative risk. 95% CI = 95% confidence interval.

ml and negative bacterial cultures, 30 day mortality was similar for patients treated with IV antibiotics for ≥72 hr and those treated with IV antibiotics for shorter periods.

Our finding that the rate of microbiologically documented bacterial co-infections from any source, including blood, lower respiratory tract, and urine, was lower in patients with PCT <0.25 ng/ml is consistent with well-established findings that pure viral infections are unlikely to increase PCT levels (*Gilbert, 2010*). Furthermore, both in the general population (*ElGohary et al., 2022*; *Azzini et al., 2020*; *Schuetz et al., 2013*; *Schuetz et al., 2017*; *Schuetz et al., 2009*; *Christ-Crain et al., 2006*) and in immunocompromised patients (*Haddad et al., 2018*; *El Haddad et al., 2018*; *Chaftari et al., 2021b*; *Chaftari et al., 2021a*), patients with low PCT levels are unlikely to have bacterial infections. PCT levels increase in patients with many types of bacterial infections, including bacterial infections of the lower respiratory tract (*Self et al., 2017*), bacterial meningitis (*Wei et al., 2016*), acute pyelo-nephritis (*Zhang et al., 2016*), spontaneous bacterial peritonitis (*Yang et al., 2015*), and bloodstream bacterial infections (*Shomali et al., 2012*). Our findings regarding PCT levels and the risk of bacterial infection are also consistent with published data on patients with COVID-19 (*So et al., 2022*; *Fabre et al., 2022*). In a recent study of patients hospitalized with COVID-19, PCT levels were higher in patients with proven bacterial co-infections: PCT level ≥0.25 ng/ml was seen in 69% of patients with proven co-infection, compared to 35% of those with low suspicion of bacterial co-infection (p<0.001) (*So et al., 2022*). The low rate of bacterial co-infection in our cancer patients with COVID-19 (about 9%) is also consistent with rates reported in the literature (*Rawson et al., 2020*; *Garcia-Vidal et al., 2021*).

Another recent study showed that PCT could be abnormally elevated in patients with COVID-19 with no evidence of pneumonia and may result in overprescribing antibiotics in such patients (*Fabre et al., 2022*). In our current study, IV antibiotics were administered to 90% of patients with PCT ≥0.25 ng/ml and 76% of patients with PCT <0.25 ng/ml (p<0.001). These high rates are similar to rates reported earlier in the pandemic, which ranged from 70% to 90% (*Wang et al., 2020*; *Chen et al., 2020*). This high rate of use of IV antibiotics in our cancer patient population could be due to the vulnerability of our immunocompromised patients. The initial PCT level may not have influenced the decision of the treating physician to initiate IV antibiotics in our frail and immunocompromised cancer patient population.

In hospitalized patients with COVID-19, PCT level ≥0.25 ng/ml was previously found to be a good predictor of oxygen supplementation, ICU admission, mechanical ventilation, and antibiotic use (*So et al., 2022*). Similarly, in our study, cancer patients with higher PCT levels were more likely to require oxygen supplementation within 72 hr of admission, be admitted in ICU, had a higher 30 day mortality rate, had a longer median duration of hospital stay, and were more likely to receive IV antibiotics.

Our data demonstrate that administering IV antibiotics beyond 72 hr in patients with PCT <0.25 ng/ml and negative bacterial cultures does not improve outcome and may be redundant. Thus, just as PCT has been used to de-escalate antibiotic use in the general population (*Schuetz et al., 2017*; *Schuetz et al., 2009*), it can be used to de-escalate antibiotic use in cancer patients with COVID-19.

Our findings that PCT <0.25 ng/ml had a negative predictive value for bacteremia, 30 day mortality, ICU admission, and IV antibiotic use >7 days are consistent with previously published data from patients with COVID-19 (*So et al., 2022*; *Heesom et al., 2020*).

The use of PCT levels to guide antibiotic therapy decisions has been important in antimicrobial stewardship outside of the COVID-19 pandemic. However, our data suggest that in cancer patients with COVID-19, if the PCT level is <0.25 ng/ml, there is low suspicion for infection, and if bacterial cultures are negative, PCT could be used as an adjunct to clinical judgment to guide de-escalation of antibiotics after 72 hr. Incorporating PCT into future algorithms for treatment of patients with COVID-19 could be cost-effective and may decrease antibiotic overuse, which is associated with unde-sirable adverse events (such as *Clostridium difficile* infection, acute kidney injury, potential allergic reactions, and loss of microbiome diversity) and contributes to the emergence of antibiotic resistance (*Azzini et al., 2020*; *Kip et al., 2015*; *Kip et al., 2018*).

Our study has limitations. First, the retrospective nature of this study may have masked confounding variables. Second, bacterial co-infections may have been overlooked given the limited face-to-face interactions with patients admitted with COVID-19 during the pandemic. Third, antimicrobials were administered empirically at the discretion of the team treating the patient. The patients were not on a defined protocol and the management of empiric antibiotic therapy as well as COVID-related

therapies were not standardized which can lead to a wide variety of practices. Hence, we have not listed the type of antibiotics that were administered either as monotherapy or in combination. Similarly, we have not reported on COVID-19 targeted therapies including immunosuppressants received such as steroids or tocilizumab. Furthermore, the study spans the period from March 2020 to June 2021 through which our knowledge of COVID has evolved, multiple variants have emerged, immunization has become available in the later part of the study period, more therapies (antivirals, monoclonal antibodies) became available, all of which could affect COVID-related mortality and outcomes. Finally, this is a single-center study, which limits the generalizability of our results.

## Conclusions

Cancer patients with COVID-19 often receive IV antibiotics despite a low rate of bacterial co-infections. Patients with low PCT levels (<0.25 ng/ml) are unlikely to have a documented bacterial infection, and they are more likely than patients with higher PCT levels to have a shorter hospital stay, shorter course of IV antibiotics, and a better overall outcome.

In cancer patients with COVID-19 and PCT <0.25 ng/ml, continuing antibiotics beyond 72 hr (or beyond when the PCT result becomes available, if antibiotic therapy has already been administered for ≥72 hr at that time) does not reduce mortality and may not have an impact on patient outcome. Hence, PCT could be used along with clinical judgment to promote antibiotic stewardship in cancer patients with COVID-19 by reducing the duration of antibiotic therapy beyond the initial empiric use of systemic antibiotics until PCT results become available.

## Acknowledgements

We thank Ms Salli Saxton, Department of Infectious Diseases, Infection Control and Employee Health, MD Anderson Cancer Center, Houston, for helping with the submission of the manuscript. We thank Stephanie Deming, Research Medical Library, MD Anderson Cancer Center, for editing the manuscript.

## Additional information

### Funding

| Funder | Grant reference number | Author |
| --- | --- | --- |
| National Cancer Institute | P30CA016672 | Issam Raad |

The funders had no role in study design, data collection and interpretation, or the decision to submit the work for publication.

### Author contributions

Hiba Dagher, Conceptualization, Writing – original draft, Writing – review and editing, acquisition of data critical review, commentary; Anne-Marie Chaftari, Conceptualization, Supervision, Writing – original draft, Writing – review and editing, oversight and leadership responsibilities, including mentorship; Patricia Mulanovich, Writing – review and editing, acquisition of data critical review, commentary; Ying Jiang, Software, Formal analysis, Visualization; Ray Hachem, Conceptualization, Writing – review and editing, oversight and leadership responsibilities, including mentorship; Alexandre E Malek, Conceptualization, Writing – review and editing, critical review, commentary, acquisiton of data; Jovan Borjan, Writing – review and editing, critical review, commentary, acquisition of data; George M Viola, Conceptualization, Writing – review and editing, critical review, commentary, acquisition of data; Issam Raad, Conceptualization, Supervision, Writing – review and editing, oversight and leadership responsibilities, including mentorship, critical review, commentary

### Author ORCIDs

Anne-Marie Chaftari ⓘ http://orcid.org/0000-0001-8097-8452

### Ethics

Our study was approved by the Institutional Review Board of MD Anderson Cancer Center, and a waiver of informed consent was obtained.

Decision letter and Author response
Decision letter https://doi.org/10.7554/eLife.81151.sa1
Author response https://doi.org/10.7554/eLife.81151.sa2

## Additional files

### Supplementary files
- MDAR checklist
- Supplementary file 1. Independent impact of PCT on outcomes by multivariable logistic regression analysis.
- Supplementary file 2. Comparing outcomes between hospitalized COVID-19 patients with different PCT values under active cancer treatment.

### Data availability
These are human subjects and we are unable to share data that contain patients' health information because of IRB restriction. We do not have the patients' consent to share their data. The study protocol, statistical analysis plan, lists of deidentified individual data, generated tables and figures will be made available upon request by qualified scientific and medical researchers for legitimate research purposes. Requests should be sent to achaftari@mdanderson.org and yijiang@mdanderson.org. Data will be available on request for 6 months from the date of publication. Investigators are invited to submit study proposal requests detailing research questions and hypotheses in order to receive access to these data. The software we used for data analysis is SAS version 9.3 (SAS Institute Inc, Cary, NC), and we have provided this information in Statistical analysis section of the manuscript.

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
