## [Editor Report]

One must appreciate the challenges of antimicrobial stewardship in an immunocompromised population. This retrospective single-institution study provides valuable support for the working hypothesis that initial procalcitonin levels might be used in cancer patients admitted with COVID-19 infection to omit, reduce, or de-escalate the need for empiric antimicrobial therapy. In the setting of a global pandemic, this is a common issue with COVID-19 patients in general, but far more difficult in a cancer patient population. The results presented are solid, however, future subgroup analysis of more specific scenarios among cancer patients with COVID-19 (e.g., neutropenia, active chemotherapy, and need for intensive care) are warranted.

---

## [Decision Letter]

**Decision letter after peer review:**

Thank you for submitting your article "Procalcitonin for Antimicrobial Stewardship Among Cancer Patients Admitted with COVID-19" for consideration by *eLife*. Your article has been reviewed by 3 peer reviewers, and the evaluation has been overseen by a Reviewing Editor and a Senior Editor. The following individuals involved in review of your submission have agreed to reveal their identity: Lisa L Dever (Reviewer #1); Vincent Yeung (Reviewer #2).

As is customary in *eLife*, the reviewers have discussed their critiques with one another. What follows below is the Reviewing Editor's edited compilation of the essential and ancillary points provided by reviewers in their critiques and in their interaction post-review. Please submit a revised version that addresses these concerns directly. Although we expect that you will address these comments in your response letter, we also need to see the corresponding revision clearly marked in the text of the manuscript. Some of the reviewers' comments may seem to be simple queries or challenges that do not prompt revisions to the text. Please keep in mind, however, that readers may have the same perspective as the reviewers. Therefore, it is essential that you attempt to amend or expand the text to clarify the narrative accordingly.

Essential revisions:

1. Please reconsider the use of the word "unnecessary" throughout the manuscript (used thrice) as it might come across as dogmatic and formulaic, but it should be contextual. In other words, there may well be instances when a patient has negative cultures and low PCT in which antibiotics are perhaps clinically indicated.

2. Please clarify certain timing issues regarding the PCT measurements, namely: A single initial PCT was used to initially sort patients. (i) Did the patients have a repeat PCT level after a few days of therapy? If PCT levels were repeated, were the results collected for the study? (ii) If the patient had a PCT <0.25 ng/ml initially, but had a subsequent value >0.25 ng/ml, were they still included in the study in the <0.25 group? (iii) If so, did these PCT changes influence clinical outcomes and antibiotic therapy duration?

3. Please clarify the criteria by which the presence of "pneumonia" is defined. Is it a clinical definition based on signs/symptoms? Or a radiographic definition based on some imaging findings? Or is the diagnosis based on documentation by the treating physician? And does this determination refer to any pneumonia, COVID-19 pneumonia, or just bacterial pneumonia?

4. Please state the antimicrobial therapy used. The institution has an antimicrobial stewardship program, but no standardized empiric therapy has been used in this setting. Therefore, the management of antimicrobial therapy has been left at the discretion of the treating physician, but the class and nature of the drugs administered is not reported here. Moreover, the authors use the terms antimicrobial, antibacterial, and antibiotic therapy interchangeably: Does antibiotic refer to antibacterial only? Finally, the use of antifungal therapy is not fully addressed, but there have evidently been more patients that had positive fungal cultures in the high PCT group. Please review the wording throughout the manuscript to ensure consistency and appropriateness.

5. Please comment on the use of COVID-19 targeted therapies, including immunosuppressants (such as steroids and tocilizumab), which may in turn increase the risk of secondary infections, but perhaps decrease the risk of mortality.

*Reviewer #1 (Recommendations for the authors):*

I have a few comments and queries for the authors to improve the manuscript.

The paper states that a single initial PCT was used to sort patients. If PCTs were repeated on patients, were the results collected for the study?

If the patient had a PCT <0.25 ng/ml initially, but had a subsequent value >0.25 ng/ml, were they still included in the study in the <0.25 group?

Were PCT levels in hemodialysis patients collected before dialysis?

Were there other patient variables associated with unreliable PCT levels?

I assume the authors' center has a robust antimicrobial stewardship program. Did this influence prescribing of antibiotics? Does the center have a protocol that PCT are collected on patients with COVID-19, or only co-infection is suspected.

The authors use antimicrobial, antibacterial and antibiotic therapy. Does antibiotic refer to antibacterial only? I would review the use in the manuscript to make sure it is consistent and appropriate. Use of antifungal therapy is not addressed, but there were more patients that had positive fungal cultures in the PCT> 0.25 group.

What criteria were used for presence of pneumonia? Imaging results? Is this bacterial pneumonia? COVID-19 pneumonia?

I would rethink the use of "unnecessary" in the discussion, conclusions and abstract. It is used 3 times in the manuscript. It comes across as dogmatic. There may be times when a patient has negative cultures and low PCT where antibiotics may be appropriate. The use of "unnecessary" is contextual.

*Reviewer #2 (Recommendations for the authors):*

– Table 1 is quite long and busy, consider shortening the list of patient characteristics.

– Would be interesting/useful to see subgroup analysis of more specific scenarios among cancer patients with COVID 19 i.e neutropenic patients, patients under active treatment, patients requiring icu level care.

*Reviewer #3 (Recommendations for the authors):*

1. Lines 94, 95, 100: I would refrain from using the word rate to describe characteristics or outcomes that do not include a time component. A better wording would be "the proportion of patients with".

2. Line 100: It might be best if the authors clarify how they defined pneumonia: Was it a clinical definition based on signs/symptoms? Or a radiographic definition based on some imaging findings? Or was it based on documentation by the treating physician? And does this refer to any pneumonia, or just bacterial pneumonia?

3. Lines 102-103: Was there a protocol of ABx de-escalation in place based on serum procalcitonin levels that prompted physicians to stop ABx early? Even as part of the institutional COVID protocol? Or was the early discontinuation of ABx solely at the discretion of the treating physician?

4. Lines 113-116: if the word count allows, I would expand a bit more on the NPV and RR values since the conclusions are based on the results of this specific analysis.

5. Lines 143-144: The authors mention the initial PCT level. Did the patients have a repeat PCT level after a few days of therapy? and if so, did this influence clinical outcomes and ABx duration?

6. Lines 147-149: The authors state that cancer patients with higher PCT levels were more likely to require O2 supplementation, be admitted to the ICU etc. Can this finding be only attributed to a bacterial coinfection?

7. Lines 154-159: These seem like a repetition of some of the results.

8. Table 1: I would split table 1 into two tables to provide better clarity for the reader:

– Baseline patient characteristics (all the way until obstructive sleep apnea).

– Characteristics of COVID hospital admission (from ANC until the end of the table).

9. Would review column width distribution in table 2.

---

## [Author Response]

Essential revisions:1. Please reconsider the use of the word "unnecessary" throughout the manuscript (used thrice) as it might come across as dogmatic and formulaic, but it should be contextual. In other words, there may well be instances when a patient has negative cultures and low PCT in which antibiotics are perhaps clinically indicated.

The use of the word “unnecessary” has been revised and toned down according to the context.

2. Please clarify certain timing issues regarding the PCT measurements, namely: A single initial PCT was used to initially sort patients. (i) Did the patients have a repeat PCT level after a few days of therapy? If PCT levels were repeated, were the results collected for the study? (ii) If the patient had a PCT <0.25 ng/ml initially, but had a subsequent value >0.25 ng/ml, were they still included in the study in the <0.25 group? (iii) If so, did these PCT changes influence clinical outcomes and antibiotic therapy duration?

Thank you for the above question. Patients may have had repeated PCT levels. For the purpose of this study we only collected the highest PCT level within the first 72 hours and classified the patients accordingly. We have clarified this point in the revised manuscript.

3. Please clarify the criteria by which the presence of "pneumonia" is defined. Is it a clinical definition based on signs/symptoms? Or a radiographic definition based on some imaging findings? Or is the diagnosis based on documentation by the treating physician? And does this determination refer to any pneumonia, COVID-19 pneumonia, or just bacterial pneumonia?

We used a radiographic definition based on imaging findings in the right clinical context in the presence of signs and symptoms to determine the presence of pneumonia by reviewing each patient’s Chest X ray and CT scan imaging report. We did not distinguish between types of pneumonia. We have added the following definition of pneumonia in the Methods section “Pneumonia was defined as an abnormal chest imaging (Chest radiograph or Computed tomography (CT) scan) in patients who present with respiratory symptoms”.

4. Please state the antimicrobial therapy used. The institution has an antimicrobial stewardship program, but no standardized empiric therapy has been used in this setting. Therefore, the management of antimicrobial therapy has been left at the discretion of the treating physician, but the class and nature of the drugs administered is not reported here. Moreover, the authors use the terms antimicrobial, antibacterial, and antibiotic therapy interchangeably: Does antibiotic refer to antibacterial only? Finally, the use of antifungal therapy is not fully addressed, but there have evidently been more patients that had positive fungal cultures in the high PCT group. Please review the wording throughout the manuscript to ensure consistency and appropriateness.

We have revised the manuscript and used antibacterial throughout for consistency. We however used the general term “antimicrobial” when referring to antimicrobial stewardship program.

We did not collect data on antifungal therapy.

We did not report on the type of antibiotics, given that this was a retrospective study, patients were not on a defined protocol, the management of empiric antibiotic therapy was not standardized but was left at the discretion of the treating physician. This was also added as a limitation in the Discussion section. The class and nature of the antibiotics used as monotherapy or in combination included: fluoroquinolones (such as levofloxacin, ciprofloxacin and moxifloxacin,), antipseudomonal/antipneumococcal β-lactams (piperacillin-tazobactam, cefepime, ceftazidime, cefazolin, ceftriaxone, meropenem, ertapenem, cefiderocol), anti-MRSA agents (vancomycin, linezolid, dalbavancin, daptomycin, clindamycin, doxycycline, linezolid, minocycline, rifampin, tigecycline, trimethoprim-sulfamethoxazole), and aminoglycoside (amikacin).

5. Please comment on the use of COVID-19 targeted therapies, including immunosuppressants (such as steroids and tocilizumab), which may in turn increase the risk of secondary infections, but perhaps decrease the risk of mortality.

We agree with the reviewer. However, we did not collect such information. This was added as a limitation in the Discussion section.

Reviewer #1 (Recommendations for the authors):I have a few comments and queries for the authors to improve the manuscript.The paper states that a single initial PCT was used to sort patients. If PCTs were repeated on patients, were the results collected for the study?If the patient had a PCT <0.25 ng/ml initially, but had a subsequent value >0.25 ng/ml, were they still included in the study in the <0.25 group?

We only collected the highest serum PCT level measured within 72 hours after admission and did not look at subsequent PCTs

Were PCT levels in hemodialysis patients collected before dialysis?

We did not collect dialysis information.

Were there other patient variables associated with unreliable PCT levels?

As shown in Tables 1 and 2 in the revised manuscript, some other variables were also significantly associated with higher PCT by univariate analysis, including chronic kidney disease, ALC<1000/ul at admission, pneumonia and oxygen supplementation within 72 hours. We thank the reviewer for this comment and following it, we performed multivariable logistic regression analyses to evaluate the independent impact of PCT >0.25 ng/ml on each outcome adjusting for possible confounders. The analyses showed that PCT>0.25 was an independent predictor of every outcome we evaluated. We have included the analyses results in Supplemental Table 1 in the revised manuscript. On the other hand, we have tried to consider all the relevant variables that could affect the outcomes being studied. We however acknowledge that because of the retrospective nature of the study (listed as a limitation), potential unmeasured confounding variables could have impacted our results.

I assume the authors' center has a robust antimicrobial stewardship program. Did this influence prescribing of antibiotics? Does the center have a protocol that PCT are collected on patients with COVID-19, or only co-infection is suspected.

PCT is ordered in our emergency center on all patients who present with a suspicion for sepsis. PCT level did not influence the physician’s prescription of antibiotics.

The authors use antimicrobial, antibacterial and antibiotic therapy. Does antibiotic refer to antibacterial only? I would review the use in the manuscript to make sure it is consistent and appropriate. Use of antifungal therapy is not addressed, but there were more patients that had positive fungal cultures in the PCT> 0.25 group.

We have revised the manuscript and used antibacterial throughout for consistency. We however used the general term “antimicrobial” when referring to antimicrobial stewardship program.

We did not collect data on antifungal therapy. We only looked at bacterial infections and antibacterial therapy.

What criteria were used for presence of pneumonia? Imaging results? Is this bacterial pneumonia? COVID-19 pneumonia?

As mentioned above, we looked at abnormal imaging results in the right clinical context and we did not distinguish between types of pneumonia.

I would rethink the use of "unnecessary" in the discussion, conclusions and abstract. It is used 3 times in the manuscript. It comes across as dogmatic. There may be times when a patient has negative cultures and low PCT where antibiotics may be appropriate. The use of "unnecessary" is contextual.

As mentioned above, the use of the word “unnecessary” has been revised and toned down according to the context and the manuscript revised accordingly.

Reviewer #2 (Recommendations for the authors):– Table 1 is quite long and busy, consider shortening the list of patient characteristics.– Would be interesting/useful to see subgroup analysis of more specific scenarios among cancer patients with COVID 19 i.e neutropenic patients, patients under active treatment, patients requiring icu level care.

We have shortened the list of patient characteristics and split table 1 into two tables (1 and 2) to provide better clarity for the reader. We adjusted the number of the following tables accordingly.

Following the reviewer comment, we performed the association analyses between PCT levels (PCT< 0.25 vs PCT ≥ 0.25) and outcomes among: (1) patients with ANC <1000 at admission; (2) patients under active cancer treatment. However, we were not able to perform such analyses among patients requiring ICU admission because we did not collect data on ICU admission at baseline. The ICU admission data we collected were an outcome data of the study. The analyses results have been added in the manuscript in Supplemental Table 2 and summarized in Results section.

Reviewer #3 (Recommendations for the authors):1. Lines 94, 95, 100: I would refrain from using the word rate to describe characteristics or outcomes that do not include a time component. A better wording would be "the proportion of patients with".

As suggested, we changed the word “rate” to “proportion of patients with” in lines 94, 95 and 100.

2. Line 100: It might be best if the authors clarify how they defined pneumonia: Was it a clinical definition based on signs/symptoms? Or a radiographic definition based on some imaging findings? Or was it based on documentation by the treating physician? And does this refer to any pneumonia, or just bacterial pneumonia?

We relied on patient imaging to determine the presence or absence of pneumonia in patients with respiratory symptoms; we did not differentiate between types of pneumonia.

3. Lines 102-103: Was there a protocol of ABx de-escalation in place based on serum procalcitonin levels that prompted physicians to stop ABx early? Even as part of the institutional COVID protocol? Or was the early discontinuation of ABx solely at the discretion of the treating physician?

The discontinuation of ABx was solely at the discretion of the treating physician.

4. Lines 113-116: if the word count allows, I would expand a bit more on the NPV and RR values since the conclusions are based on the results of this specific analysis.

We agree and thank the reviewer for the advice. We have expanded our description on NPV and RR data by providing details in the Results section.

5. Lines 143-144: The authors mention the initial PCT level. Did the patients have a repeat PCT level after a few days of therapy? and if so, did this influence clinical outcomes and ABx duration?

For this study we only collected and analyzed the highest initial PCT level within the first 72 hours and not subsequent PCT levels.

6. Lines 147-149: The authors state that cancer patients with higher PCT levels were more likely to require O2 supplementation, be admitted to the ICU etc. Can this finding be only attributed to a bacterial coinfection?

We performed Cochran-Mantel-Haenszel (CMH) test to evaluate the association between PCT level and oxygen supplementation requirement while adjusting for bacterial coinfection and found that their association after the adjustment remains significant (p=0.002). Similarly, the association between PCT and ICU admission remains significant after adjustment for bacterial coinfection (p<.0001) by a CMH test. In addition, we performed multivariable logistic regression analysis on each outcome and found that PCT>=0.25 was independently associated with each outcome we evaluated after adjusting for possible confounders. The analyses results were added in Supplemental Table 1 in the revised manuscript.

7. Lines 154-159: These seem like a repetition of some of the results.

This paragraph was deleted.

8. Table 1: I would split table 1 into two tables to provide better clarity for the reader:– Baseline patient characteristics (all the way until obstructive sleep apnea).– Characteristics of COVID hospital admission (from ANC until the end of the table).

Thank you for your suggestion. We have split the table into Table 1 and Table 2 according to your suggestion and edited the number of the other tables accordingly.

9. Would review column width distribution in table 2.

We have reformatted old tables 2 and 3 (that are now tables 3 and4).